# A mechanistic reinterpretation of fast inactivation in voltage-gated Na$^+$ channels

Yichen Liu [1], Carlos A. Z. Bassetto Jr [2], Bernardo I. Pinto[2] & Francisco Bezanilla [2,3] ✉

The hinged-lid model was long accepted as the canonical model for fast inactivation in Nav channels. It predicts that the hydrophobic IFM motif acts intracellularly as the gating particle that binds and occludes the pore during fast inactivation. However, the observation in recent high-resolution structures that the bound IFM motif is located far from the pore, contradicts this preconception. Here, we provide a mechanistic reinterpretation of fast inactivation based on structural analysis and ionic/gating current measurements. We demonstrate that in Nav1.4 the final inactivation gate is comprised of two hydrophobic rings at the bottom of S6 helices. These rings function in series and close downstream of IFM binding. Reducing the volume of the sidechain in both rings leads to a partially conductive, leaky inactivated state and decreases the selectivity for Na$^+$ ion. Altogether, we present an alternative molecular framework to describe fast inactivation.

Voltage-gated sodium (Nav) channels function as the main drivers of the depolarization phase of action potentials. As such, they are ubiquitously found in a wide range of excitable tissues, playing fundamental roles in many vital physiological functions, such as skeletal muscle contraction, heart beating, and neuronal impulse generation and propagation[1]. Consisting of one long protein chain, mammalian Nav channels are arranged into four similar, yet not identical domains (DI to DIV). Each domain, formed by 6 transmembrane helices (S1 to S6), is then organized pseudo-symmetrically into the voltage-sensing subdomain (VSD, S1 to S4) and the pore-forming subdomain (PD, S5 to S6)[2]. Milliseconds after activation, Nav channels rapidly enter into a non-conductive state, the fast inactivated state, and only recover upon repolarization[3–6]. Fast inactivation serves as a negative feedback switch, responsible for the attenuation of inward current during action potential. Subtle abnormalities in fast inactivation can lead to severe pathological states[7–9]. These diseases, though sharing similar etiology, can manifest in many different types of tissues and show a wide range of symptoms due to the prevalence and diversity of Nav channels[10]. For instance, skeletal muscle hyperkalemic periodic paralysis[11,12], long QT syndrome in the heart[13], and inherited epilepsy in the brain[14] can all arise from abnormal Nav fast inactivation.

While examining the ionic and gating current from Nav channels in the giant squid axon, Armstrong and Bezanilla formulated the ball and chain model[15,16], where a positively charged "inactivation ball", tethered by a flexible "chain" to the intracellular domain of the channel, directly blocks the permeation pathway. The ball and chain were thought to be part of an autoinhibitory process on the intracellular face of the channel since fast inactivation is keenly susceptible to proteolytic action in internally perfused axons[17]. Site-directed mutagenesis experiments led to the identification of a triad of hydrophobic residues (Isoleucine-Phenylalanine-Methionine, the IFM motif) in the linker between DIII and DIV that is necessary for fast inactivation[18]. Modifying the original "ball and chain" model, West et al. proposed a hinged-lid model, where the IFM motif was predicted to be the inactivation particle that physically blocks the pore during fast inactivation, acting as the inactivation gate. Many reports have subsequently supported this model, and the hinged-lid model was essentially accepted as the canonical model for fast inactivation in Nav channels[19–21] until the advent of the cryoEM "resolution revolution".

Upon the first mammalian Nav channel structure[22], it became clear that the IFM motif docked into a hydrophobic pocket far away from the pore rather than physically occluding the pore, in contrast to the predictions of the canonical hinged-lid model. Since then, numerous

[1]Department of Neurobiology, University of Chicago, Chicago, IL, USA. [2]Department of Biochemistry and Molecular Biology, University of Chicago, Chicago, IL, USA. [3]Centro Interdisciplinario de Neurociencias de Valparaíso, Valparaíso, Chile. ✉e-mail: fbezanilla@uchicago.edu

structural reports of different isoforms of Nav channels all confirm this original observation[23–29]. And while the IFM motif does not seem to serve as the fast inactivation gate, it is clearly necessary for the fast inactivation process[18]. As a result, the precise molecular determinants of fast inactivation gate have, once again, become an open question.

Here, we take advantage of the available structural data and test directly the hypothesis suggesting that, rather than the IFM motif, fast inactivation in Navs is associated with large residues at the intracellular end of pore-forming helices S6. Combining ionic and gating current measurements, we demonstrate and characterize an S6-located double ring of hydrophobic residues in DIII and DIV. Through site-directed mutagenesis, we show that the fast inactivation gate is formed by two layers of bulky, hydrophobic residues, whose effectiveness as an inactivation gate is dependent on side chain volume. Once reduced, smaller side chain substitutions lead to a leaky inactivated state. Equivalent residues in DII, on the other hand, seem to be involved in the coupling between VSD activation and PD opening instead. Surprisingly, our experiments point to a previously overlooked coupling pathway between the bottom of S6 and the selectivity filter (SF). Altogether, our results identify an alternative candidate and mechanism for describing the fast inactivation gate in Nav channels, highlighting the importance of the intracellular (C-terminal) end of S6 for both activation and fast inactivation.

## Results

### Structural analysis reveals a two-tier hydrophobic barrier at the bottom of S6

Starting with the observation that the IFM motif is located far from the pore in the putative inactivated state (Fig. 1B)[28,30,32], we reasoned that large, hydrophobic residues residing at the intracellular end of S6 might play a dual role participating both in activation and fast inactivation gating. Close examination of the electron density maps in the available Nav channel structures shows that in most cases, non-protein densities can be spotted inside the pore, spanning across the bundle crossing region into the inner cavity[22,32,33]. These molecules are sometimes unidentified. However, in many cases assigned to be detergent

molecules, clouding any interpretation regarding their functional state. Here we have focused our analyses on two structures that do not present such non-protein densities in the pore: NavPas (PDB:6A95[30], Fig. 1A) and Nav1.7 M11 (PDB:7XVF[28], referred to as Nav1.7 from now on, Fig. 1B). Given the fact that both structures are determined at 0 mV, and the DIV VSD is in the up conformation, it is reasonable to assume that the pore is in an inactivated state. We calculated the pore profile in NavPas and Nav1.7 (HOLE) as the radius along the permeation pathway (graphs in Fig. 1A, B). These profiles led to easy identification of the narrowest part of the pore in each case. While NavPas and Nav1.7 showed very similar pore profiles at the intracellular end, the stretch along the pore that forms the narrowest part was much larger in NavPas and Nav1.7 than structures with non-proteinaceous densities inside the pore, for instance, in the case of Nav1.5 structure (Supplementary Fig. 1). The narrowest region in NavPas and Nav1.7 was composed by two layers of hydrophobic residues organized in a diamond shape facing towards the pore at the bottom of S6, seen as two minima in the radius profiles (Fig. 1A, B), located one α-helix turn away from each other. An early indication of the feasibility of our hypothesis is related to the fact that all the residues identified were hydrophobic, and 7 of them were also bulky. Additionally, the 8 residues (two from each domain) identified in NavPas and Nav1.7 are highly conserved among different Nav isoforms (Supplementary Fig. 2). Based on our structural analysis, we hypothesize that these S6 residues form a two-tier hydrophobic barrier in the fast inactivated state and serve as the fast inactivation gate in Nav channels.

### Double alanine mutation in DIII S6 produces a conductive leaky inactivated state

After identifying residues that could potentially form the fast inactivation gate at S6, we predicted that by reducing the volume of the involved side chains in BOTH layers simultaneously, we could widen the two-tier barrier in S6 and partially open the fast inactivation gate. We first tested DIII and DIV due to their known role in fast inactivation and the fact that the IFM motif is spatially closer to these two domains (Fig. 1B)[34–36]. In DIII, isoleucine I1284 and I1288 were identified based on

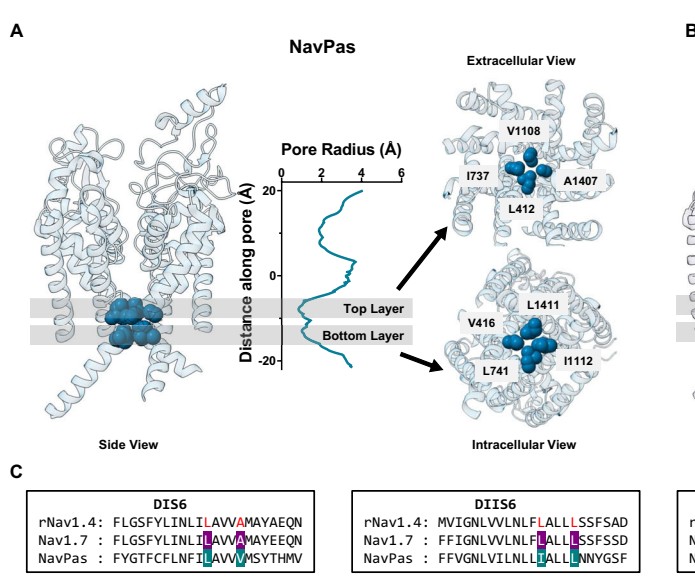

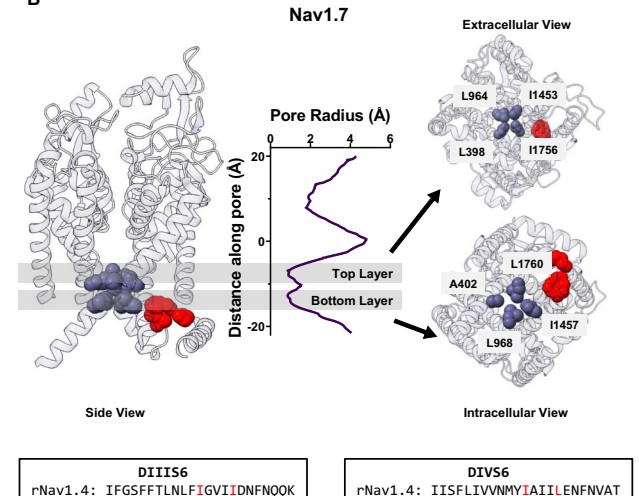

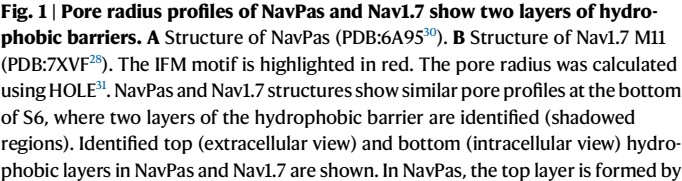

**Fig. 1 | Pore radius profiles of NavPas and Nav1.7 show two layers of hydrophobic barriers. A** Structure of NavPas (PDB:6A95[30]). **B** Structure of Nav1.7 M11 (PDB:7XVF[28]). The IFM motif is highlighted in red. The pore radius was calculated using HOLE[31]. NavPas and Nav1.7 structures show similar pore profiles at the bottom of S6, where two layers of the hydrophobic barrier are identified (shadowed regions). Identified top (extracellular view) and bottom (intracellular view) hydrophobic layers in NavPas and Nav1.7 are shown. In NavPas, the top layer is formed by residues L412 (DI), I737 (DII), V1108 (DIII), A1407 (DIV), and the bottom layer by V416 (DI), L741 (DII), I1112 (DIII), L1411 (DIV). In Nav1.7, the top layer is formed by residues L398 (DI), L964 (DII), I1453 (DIII), I1756 (DIV), and the bottom layer by A402 (DI), L968 (DII), I1457 (DIII), L1760 (DIV). C Sequence alignments of the S6 region among rNav1.4, Nav1.7, and NavPas. The residues identified to form the narrowest part of the channel were highlighted in cyan for NavPas and purple for Nav1.7, and they are colored in red for rNav1.4.

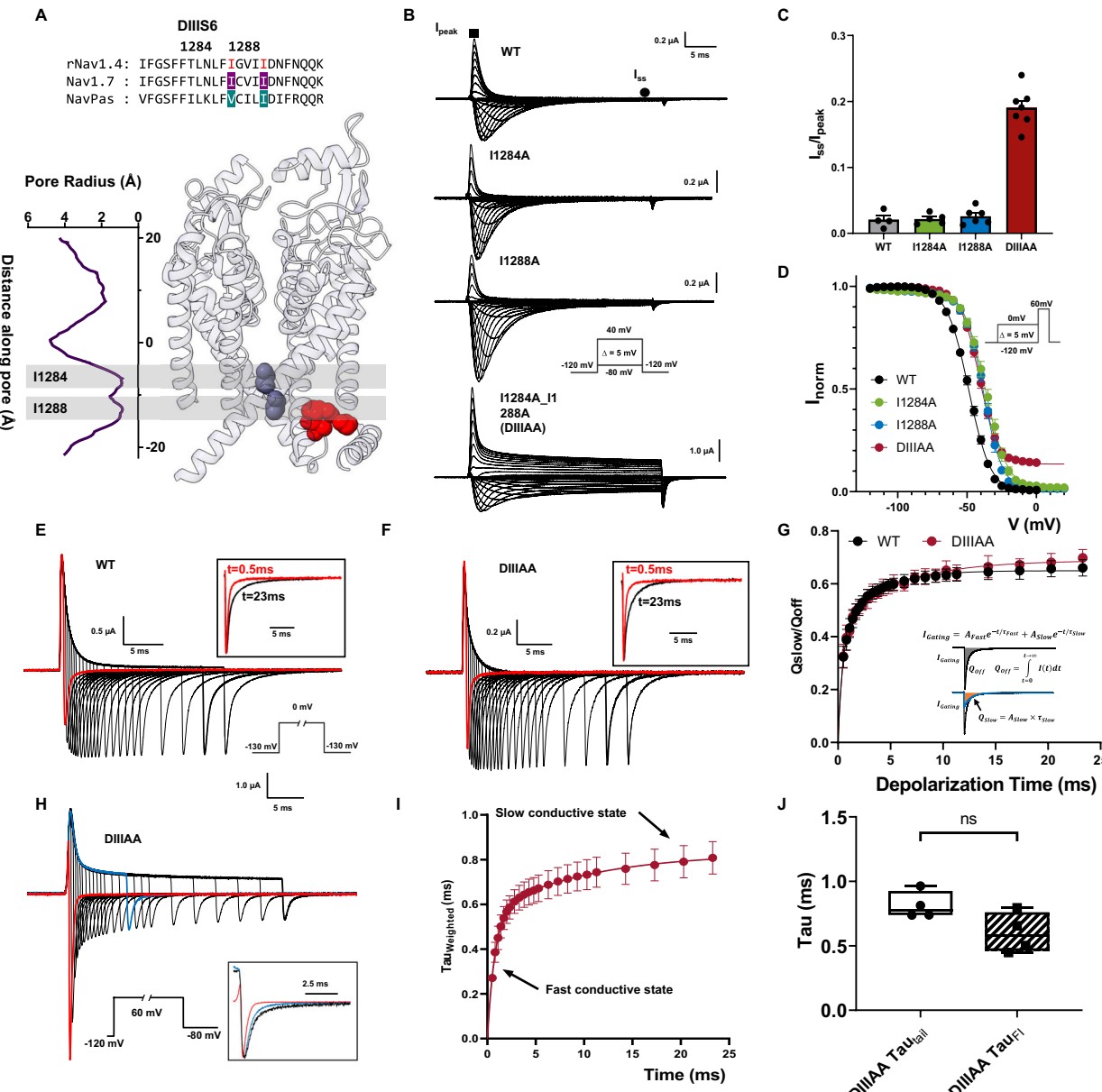

**Fig. 2 | Double alanine mutation of the identified hydrophobic residues in DIII S6 creates a leaky inactivated state in rNav1.4. A** Sequence alignment of DIII S6 showing identified residues (I1284 and I1288) and their positions in the Nav1.7 structure. **B** Representative ionic traces for the wild-type (WT), single (I1284A and I1288A), and double alanine (I1284A_I1288A, named DIIIAA) mutations of the identified residues on rNav1.4. The ionic conditions used were 57.5 mM $Na^+$ outside and 12 mM $Na^+$ inside. Inset shows the voltage protocol. WT, $N = 4$; I1284A $N = 5$; I1288A, $N = 6$; DIIIAA, $N = 7$. **C** Ratio of steady state current ($I_{ss}$, black circle) at the end of 30 ms depolarization over the peak current ($I_{peak}$, black square) taken at +60 mV. **D** Voltage-dependence of inactivation ($h−$infinity curve) for WT (black), I1284A (green), I1288A (blue), and DIIIAA (red). Inset shows the voltage protocol. The lines represent the fit to a two-state model (Eq. 1). WT, $N = 8$; I1284A, $N = 6$, I1288A, $N = 6$; DIIIAA, $N = 4$. **E, F** Representative traces of gating current for WT (**E**) and DIIIAA (**F**). Inset shows the voltage protocol. The black square top inset shows the comparison of the normalized gating current after 0.5 ms and 23 ms depolarization pulse. **G** Fraction of immobilized charge vs. depolarization time for WT (black) and DIIIAA mutant (red). The inset illustrates the method used for the determination of the off-gating components using exponential fitting (Detailed in methods). WT, $N = 4$; DIIIAA, $N = 4$. **H** DIIIAA currents in response to a depolarized voltage step to +60 mV with variable duration followed by a hyperpolarized voltage step to −80 mV. The inset shows the normalized tail currents at 3 depolarization times. **I** Weighted tail currents time constant vs. depolarization time for DIIIAA mutant, fitted with two-component exponential association (Details in methods), $N = 4$. **J** Comparison of the fast time constant of the slowing down of the tail current and the fast time constant of fast inactivation at +60 mV, $N = 4$ for both measurements. No significant difference was found (unpaired t-test with two-sided Welch's correction, $p > 0.5$). The whisk shows the maxima and minima, the center shows the mean, and the box shows the 25th to 75th percentiles. All data are shown as mean ± SEM.

the sequence alignment in rNav1.4, our model Nav channel (Fig. 2A). Single alanine mutations on either position (I1284A or I1288A) yielded currents similar to wild type (WT) channel (Fig. 2B). When both residues were mutated to alanine simultaneously (mutant DIIIAA), the significant steady state current was observed (Fig. 2B, bottom traces). At +60 mV, this steady state current corresponds to around 20% of the peak current, which clearly differs from the negligible amounts (less than 3%) in the WT as well as in the I1284A and I1288A mutants (Fig. 2C). A small right shift, around 10 mV, can be seen in the voltage dependence of the fast inactivation (h-infinity curve) in I1284A, I1288A, and DIIIAA (Fig. 2D, Supplementary Table 1). The steady state current was present exclusively in the double mutant, which cannot be accounted

for by a combination of the effects of the two single mutants since neither of them showed a steady state current. But this is expected if the two-tier barrier is acting as a steric hindrance in series, as is observed in the pore radius measurement where two layers of hydrophobic barriers block the pore at the bottom of S6, and the permeation pathway could only be widened by reducing both residues forming the barrier.

To characterize the conformational changes along the fast inactivation pathway, we measured gating charge immobilization in DIIIAA. As Nav channels enter fast inactivation, VSDs in DIII and DIV get trapped in the up conformation and become immobilized. As a result, upon hyperpolarization, VSDs in DIII and DIV move after the channels transitioned out of the fast inactivated state, manifesting as a slow component in the off-gating current[16]. Therefore, the amount of immobilized charge and the time course of immobilization provide a direct measurement of conformational changes along the fast inactivation pathway, independent of ionic current. In WT, almost all channels are inactivated milliseconds after depolarization (Fig. 2B). Consistent with this, a second slow kinetic component develops in the off-gating current milliseconds after depolarization (Fig. 2E). Despite the presence of steady state ionic current (Fig. 2B, C), a similar pattern of charge immobilization was observed in the off-gating currents of DIIIAA (Fig. 2F, G). In WT and DIIIAA about 60% of the charge was immobilized, and both constructs shared a similar time course (Fig. 2G). This result suggests that DIIIAA channels transition completely into the fast inactivated state, and the steady current observed cannot be explained, assuming a subpopulation of channels is unable to enter the inactivated state. The fact that the time course of immobilization is about the same between WT and DIIIAA, indicates that the

rate constants of transitioning in and out of fast inactivation were not significantly affected by the double alanine mutation. This is further supported by the essentially identical kinetics of the ionic current inactivation in DIIIAA and WT (Supplementary Fig. 3). Thus, the kinetic pathway from the open state to the fast inactivated state was not dramatically altered in DIIIAA, consistent with the charge immobilization results.

Upon short voltage depolarization, tail currents from DIIIAA mutant were fast and exhibited a single time constant of -0.3 ms. But with longer depolarizations, the tails slowed down considerably (Fig. 2H). This was evidenced by the appearance of a second slow component, reaching a weighed time constant close to 0.8 ms, pointing to the existence of two distinct conductive states (Fig. 2I). The slowing down of the tail kinetics followed the fast time constant of the apparent fast inactivation in DIIIAA (Fig. 2J), indicating that as the fast inactivation proceeds, the channel transition from one conductive state to the other.

Combining all our results, it is clear that the steady state current seen in DIIIAA is not a result of incomplete fast inactivation. All the channels still transition into the fast inactivated state. However, due to the double alanine mutation, the fast inactivation gate becomes partially open, and the final fast inactivated state remains conductive, manifested as a second conductive state, a leaky inactivated state.

### The leaky inactivated state is less selective for Na$^+$ ion
In certain voltage ranges, we noticed that the ionic currents went from inward to outward during the depolarizing pulse in DIIIAA. This was shown as a fast inward and a steady state outward component (Fig. 3A and Supplementary Fig. 4A–C). Since both current components are

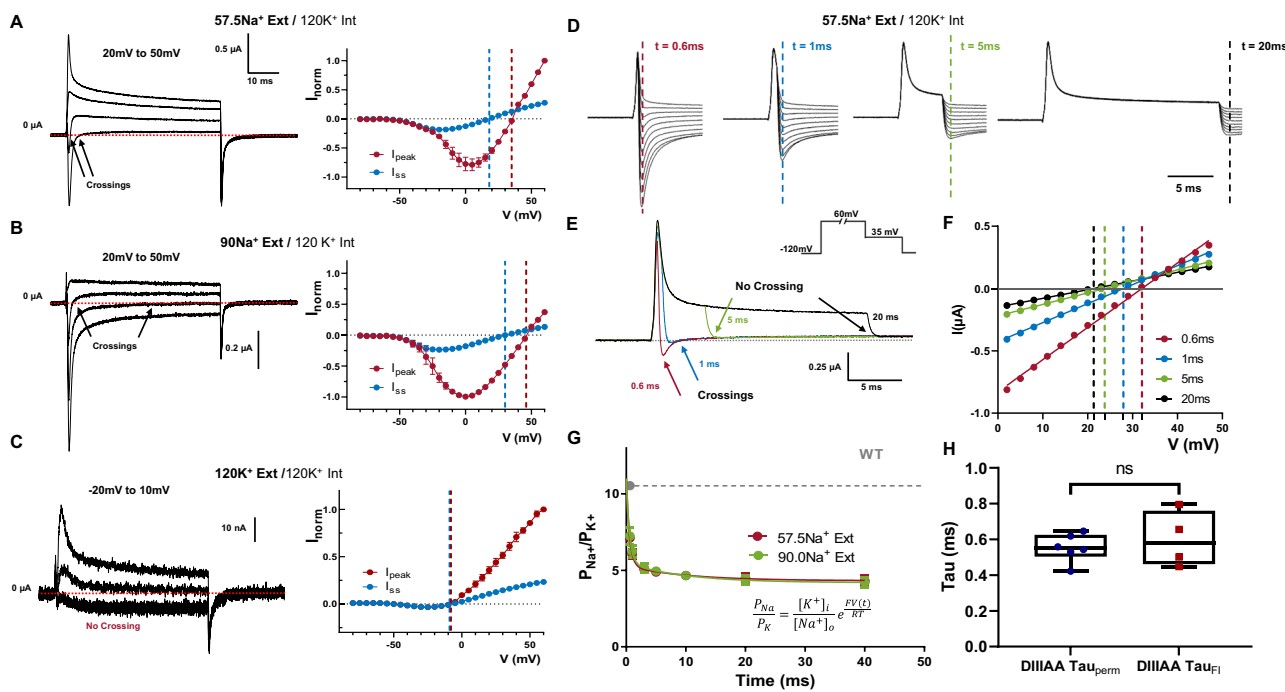

**Fig. 3 | Time-dependent selectivity changes in DIIIAA.** DIIIAA currents at different voltages (range shown on top) using 57.5 mM external Na$^+$ (**A**), $N = 3$, 90 mM external Na$^+$ (**B**), $N = 3$ or 120 mM external K$^+$ (**C**), $N = 2$ with 120 mM internal K$^+$; on the right $I-V$ curves for the peak (red) and steady state (blue) current. The reversal potential for each component is denoted with a corresponding dashed line. **D** Instantaneous $I-V$ protocol obtained at different times during depolarization. **E** Example of the changes in direction for the tail currents. **F** Example instantaneous $I-V$ curves after a 0.6 (red), 1 (blue), 5 (green), and 20 (black) milliseconds depolarization times of a single oocyte. The solid lines represent a linear fit used to obtain the reversal potential (indicated by vertical dashed lines). **G** Relative Na$^+$/K$^+$

permeability vs. time for DIIIAA under different ionic conditions (red, 57.5 mM Na$^+$ external and green, $N = 4$, 90 mM Na$^+$ external concentration, $N = 3$). Internal solution 120 mM K$^+$. The dashed line shows the WT permeability, and solid lines represent a two-component exponential fit to obtain the time constants. **H** Fast time constants for the change in selectivity (DIIIAA Tau$_{perm}$, $N = 6$) and fast inactivation time constants obtained from the ionic current at +60 mV (DIIIAA Tau$_{FI}$, $N = 4$) show no significant difference (unpaired $t$-test with two-sided Welch's correction, $p > 0.5$). The whisk shows the maxima and minima, the center shows the mean, and the box shows the 25th to 75th percentiles. Data are shown as mean ± SEM.

blocked by TTX (Supplementary Fig. 4D), both are permeating through the pore, and the selectivity of the channel might be changing during the voltage pulse. Indeed, the reversal potential for the peak current, which is related to the first component, is significantly more positive than the one for the steady state current, related to the second component in bi-ionic environment (Fig. 3A). When external $Na^+$ was increased from 57.5 to 90 mM both components were shifted to more depolarized voltages: from 35 to 46 mV for the peak component and from 18 to 30 mV for the steady state component (Fig. 3A, B). When only $K^+$ is used as a permeant ion, there is no crossing in the ionic currents, and the reversal potential for the peak and steady state are the same (Fig. 3C). Once $K^+$ was exchanged by $Na^+$ in the external solution, the crossing reappears, showing that this process is reversible (Supplementary Fig. 5). Therefore, surprisingly enough, the relative permeability between $Na^+$ and $K^+$ is changing during the time course of depolarization in DIIIAA.

To further explore this change in selectivity, we measured the relative permeability at different time points during a depolarizing voltage pulse (60 mV). This was done by measuring the instantaneous $I-V$ curve at different times during depolarization, computing the reversal potential as a function of time, and using the GHK equation (Eq.(6)) as a framework to analyze the data (Fig. 3D–F, "Methods"). Under bi-ionic conditions, we can calculate the relative permeability between $Na^+$ and $K^+$ ions ($P_{Na}/P_K$) from the reversal potential measurements. At the beginning of the pulse, the selectivity of DIIIAA toward $Na^+$ was high, similar to the WT ($P_{Na}/P_K \approx 11$), but almost three times less selective later in the pulse ($P_{Na}/P_K \approx 4$) (Fig. 3G). The fast time course of the permeability changes also followed the time course of the apparent fast inactivation, indicating that the conformational changes associated with fast inactivation might also trigger a change in selectivity (Fig. 3H). These results point to the existence of an allosteric coupling pathway between S6 and the SF. Independent of the underlying coupling mechanism, these results show unequivocally that the bulky hydrophobic residues identified form part of the inactivation gate.

### The leaky inactivated state can be accessed from a closed state and can be prevented by removing fast inactivation in DIIIAA

So far, we have found no evidence to suggest that the reduction in the side chain volume in DIIIAA has any effect on the leakiness of the closed state at rest. One interesting feature of the present results is that at hyperpolarized voltages (from −55 mV to −30 mV), the DIIIAA mutant displays slower ionic currents kinetics but no apparent inactivation, a phenomenon not observed for the single mutants and WT (Fig. 4A). When compared to WT, the voltage dependence of activation ($G-V$ curve) of DIIIAA was shifted −8.5 mV to the left (calculated at the steady state), whereas the two single alanine mutants were right-shifted (+15.2 mV for I1284A; +9.6 mV for I1288A) (Fig. 4B, Supplementary Table 2).

Despite this, the voltage dependence of the voltage sensor movement ($Q-V$ curve) showed a shift to the right in DIIIAA (+8.5 mV) when compared to the WT (Fig. 4C, D–Supplementary Table 3). On the surface, this suggests a possible increase in the coupling between the VSD and PD for DIIIAA. However, one alternative explanation for this paradoxical result was that the leaky inactivated state present in DIIIAA can be accessed at less depolarized voltages before activation, shifting the G-V curves to the left. Since fast inactivation in Nav channels precedes activation in voltage[35,37], this would explain the early component in the current as flowing through the leaky inactivated state.

In order to test our hypothesis, we introduced the F1304Q mutation in the IFM motif in both WT (IQM) and DIIIAA (IQM_DIIIAA) to remove fast inactivation. Within the previously accepted hinged-lid model, IQM was supposed to remove fast inactivation by preventing the binding of IFM to its ultimate receptor in the open pore. In the

current framework where we demonstrate that the residues at S6 form the fast inactivation gate, we interpret the IFM motif, instead of being the final effector, acts more likely as a transducer that couples the VSD movement in DIV, which triggers the fast inactivation, to the pore where the inactivation gate ultimately closes. By mutating F1304 to Q, the energetic barrier becomes so high that, effectively, the conformational changes associated with fast inactivation are terminated before reaching the pore. By preventing fast inactivation, we expect to remove the features associated with the leaky inactivated state, namely: the slow component on the tail currents, the change in selectivity, and the shift in the G-V curve. In WT, IQM removes most of the fast inactivation without significant shifts in the $G-V$ curve (Fig. 4E, F, Supplementary Table 2). In contrast, in IQM_DIIIAA, the $G-V$ curve was shifted to the right (+9.6 mV) compared to the WT, comparable to the single mutants, and to the shift in the DIIIAA $Q-V$ curve. We saw no change in selectivity and tail currents kinetics did not change with the depolarization time (Fig. 4G–I). Since the IQM mutation itself does not change the G-V curve and removes inactivation, our results indicate that: (i) the leaky inactivated state appears downstream of IFM binding; (ii) the apparent shift seen in DIIIAA $G-V$ curve was likely due to the channel entering into the leaky fast inactivation state from a closed state which contributed to the early current seen at hyperpolarized voltages; and (iii) the change in selectivity is only apparent as channels transition into the leaky inactivated state. Additionally, this result implies that the high energy barrier associated with the sharp reduction in conductance in the closed state is likely linked to a separate set of S6 residues.

### Residues identified at DIV S6 are also part of the inactivation gate

DIV, similar to DIII, has been shown to have specialized roles in fast inactivation. We identified two large residues at the bottom of DIV S6 based on the sequence alignment, I1587 and L1591 (Fig. 5A). When I1587 and L1591 residues were mutated to alanine simultaneously (DIVAA), significant steady state current was observed across all voltages tested (around 13% at 60 mV) (Fig. 5B–D). However, single alanine mutations (I1587A, L1591A) did not show a significant change in fast inactivation (Fig. 5B–D−Supplementary Table 1).

Despite the steady state current in DIVAA, the amount of charge immobilization is similar to the WT and DIIIAA (Fig. 5E, F). Furthermore, in DIVAA, the apparent fast inactivation process had a similar time course as the WT and DIIIAA (Supplementary Fig. 6). These results show that DIVAA channels can also transition into the fast inactivated state but, similarly to DIIIAA, the fast inactivation gate is partially open, creating a leaky inactivated state. Consistently, two components of current with different reversal potentials are observed in DIVAA (Fig. 5G, H and Supplementary Fig. 6). After determining the relative permeability between $Na^+$ and $K^+$ as a function of time, we observed a decrease in selectivity in DIVAA. At the onset of the pulse, the selectivity of DIVAA towards $Na^+$ was high ($P_{Na}/P_K \approx 13$), but later in the pulse, the relative permeability for Na decreased ($P_{Na}/P_K \approx 2$) (Fig. 5I). The change in selectivity follows the kinetics of inactivation as evidenced by their similar time constants (Fig. 5J). The gating charge movement and the activation process are not drastically affected by the double alanine mutation DIVAA (Supplementary Fig. 6). Based on the many parallels between DIIIAA and DIVAA, we believed that the effects produced by the double alanine mutation in DIII S6 and DIV S6 share the same underlying mechanism and the identified hydrophobic residues in DIII together with the ones in DIV are part of the fast inactivation gate.

To further demonstrate the shared role of DIII and DIV S6 as part of the inactivation gate, we mutated all four residues identified to alanine (DIII_IVAA, Fig. 6A). The effect of the quadruple alanine mutation was drastic. This mutant exacerbated the effects seen in DIIIAA and DIVAA alone. Firstly, a significant steady state current was

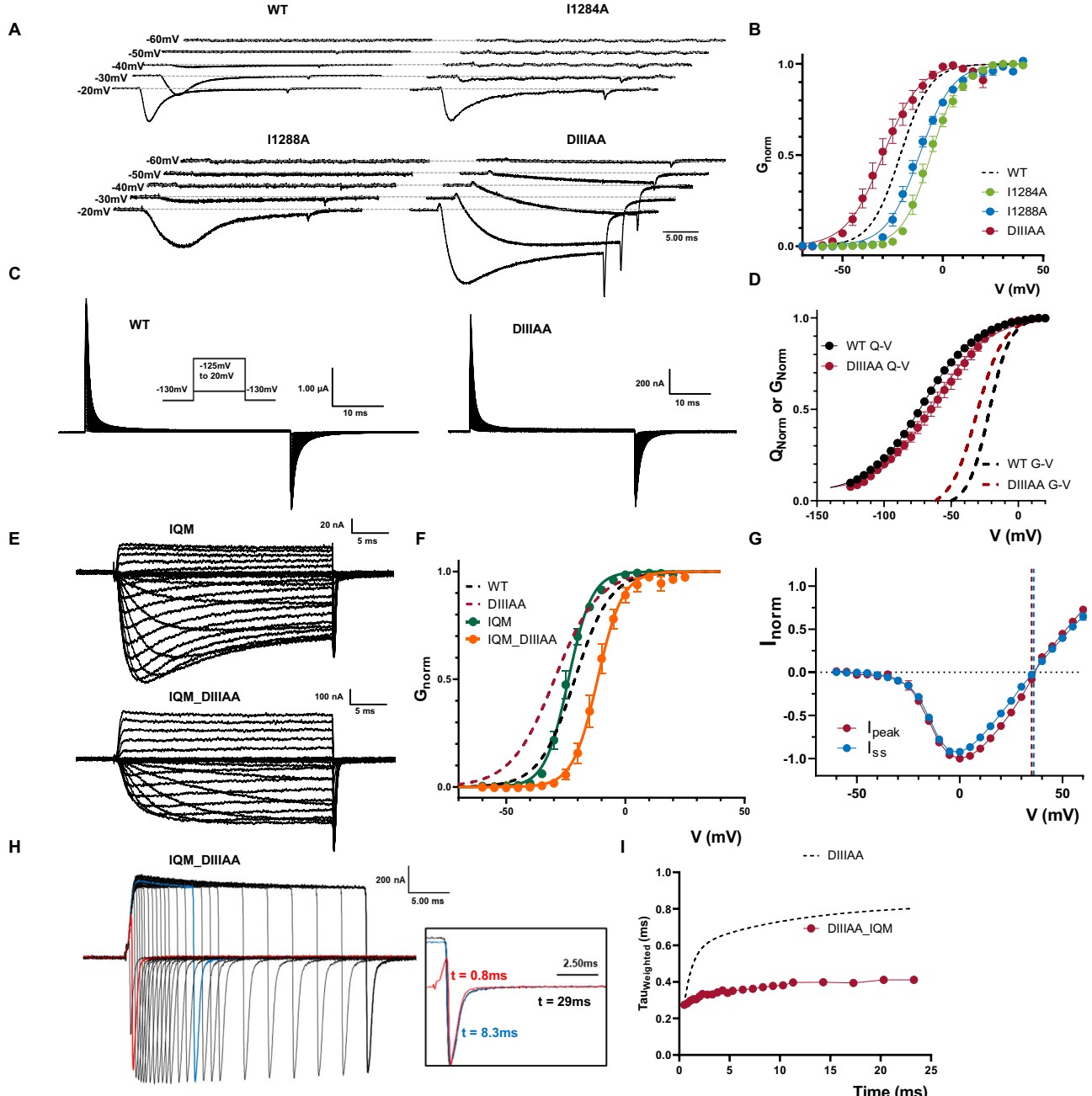

**Fig. 4 | Preventing IFM binding avoids the DIIIAA phenotypes. A** Example traces of WT, I1284A, I1288A, and DIIIAA. Early current in DIIIAA developed in more hyperpolarized voltages and showed no clear fast inactivation. **B** G–V curves for WT (black dashed line), I1284A (green), I1288A (blue), and DIIIAA (red). G–V curves were calculated from the peak for WT, I1284A and I1288A, whereas for DIIIAA it was calculated from the steady state currents (WT, $N = 4$; I1284A, $N = 4$; I1288A. $N = 6$; DIIIAA, $N = 4$). **C** Representative gating current traces for WT and DIIIAA. Inset is the voltage protocol. **D** Q–V curves for WT (black, $N = 6$) and DIIIAA (red, $N = 6$). The dashed lines indicate the corresponding G–V curves. **E** Representative ionic current traces for IQM and IQM_DIIIAA. **F** G–V curves for IQM (green, $N = 4$) and IQM_DIIIAA

(orange, $N = 4$). Dashed lines show the WT and DIIIAA for comparison. **G** IQM_DIIIAA I–V curves for the peak (red $N = 4$) and steady state (blue, $N = 4$) current. The reversal potential for each component is denoted with a corresponding dashed line. **H** IQM_DIIIAA currents in response to a depolarized voltage step to +60 mV with variable duration followed by a hyperpolarized voltage step to −80 mV. The inset shows the normalized tail currents at 3 different depolarization times. **I** Weighted tail currents time constant vs. depolarization time for IQM_DIIIAA (red points, $N = 5$) compared to DIIIAA (dashed line, $N = 4$). Data are shown as mean ± SEM.

seen (Fig. 6B, C). Secondly, the ionic selectivity became severely impaired, as evidenced by the fast inward and steady state outward component at some voltages (Fig. 6B, D). The $P_{Na}/P_K$ at the onset of the pulse was already ~2.4 and at steady state ~1.5 (Fig. 6E). These results further demonstrate that all four residues are part of the fast inactivation gate and reinforce the hypothesis of a coupling between the bottom of S6 region and the SF.

## Residues identified in DI and DII S6 contribute to the channel behavior differently

The effects of alanine mutations in DI and DII were drastically different from DIII and DIV. In DI S6, out of the two identified residues, only one residue was large and hydrophobic (L437), and the second identified residue was an alanine (A441, Fig. 7A). Therefore, we only tested a single alanine mutation (L437, DIA). In DIA, a small amount

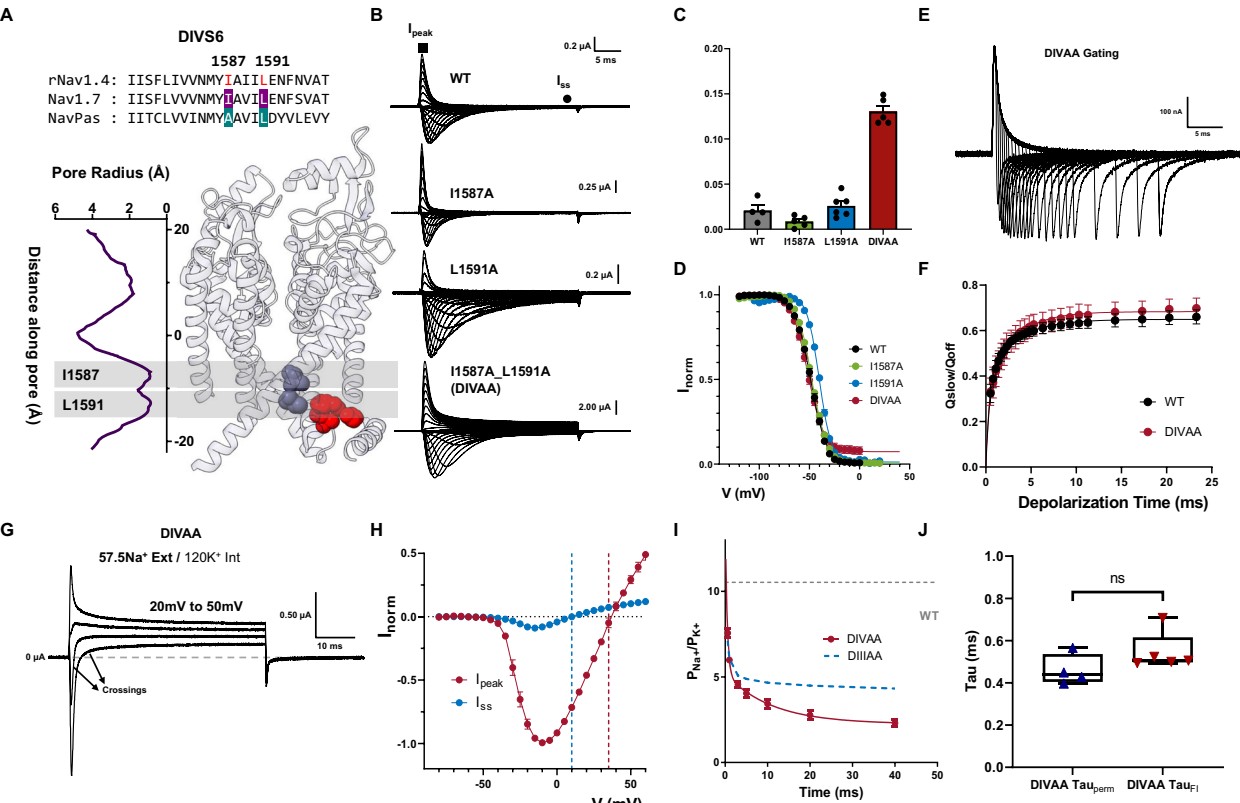

**Fig. 5 | Double Alanine mutations in DIV also produced a leaky inactivated state. A** Sequence alignment of DIV S6 showing identified residues (I1587 and L1591, highlighted in red) and their respective positions in the Nav1.7 structure. **B** Representative ionic traces for single (I1587A and L1591A) and double alanine (I1587A_L1591A−DIVAA) mutations. The ionic conditions used were 57.5 mM $Na^+$ outside and 12 mM $Na^+$ inside. **C** Ratio of steady state current ($I_{ss}$) at the end of depolarization over the peak current ($I_{peak}$) taken at +60 mV. WT, $N = 4$; I1587A, $N = 5$; L1591A, $N = 6$; DIVAA, $N = 5$. **D** Voltage-dependence of inactivation ($h$−infinity curve) for WT (black, $N = 8$), I1587A (green, $N = 5$), L1591A (blue, $N = 5$), and DIVAA (red, $N = 8$). **E** Representative gating current traces for charge immobilization measurements for DIVAA. **F** Fraction of immobilized charge vs. depolarization time for WT (black, $N = 4$) and DIVAA mutant (red, $N = 7$). **G** Representative ionic currents

trace in bi-ionic environment. **H** $I$−$V$ curves for the peak (red, $N = 5$) and steady state (blue, $N = 5$) current for DIVAA. The reversal potential for each component is denoted with the corresponding dashed line. **I** Relative $Na^+$/$P_{K+}$ permeability vs. time for DIVAA (57.5 mM $Na^+$ external and 120 mM $K^+$ internal, $N = 4$). Dashed gray and blue lines show the WT and DIIIAA permeability, respectively. The solid red line represents an exponential fit to obtain the time constant. **J** Fast time constants for the change in selectivity (DIVAA $Tau_{perm}$, $N = 4$) and fast inactivation time constant obtained from the ionic current at +60 mV (DIVAA $Tau_{FI}$, $N = 5$) show no significant difference (unpaired $t$-test with two-sided Welch's correction, $p > 0.5$). The whisk shows the maxima and minima, the center shows the mean, and the box shows the 25th to 75th percentiles. Data are shown as mean ± SEM.

of steady state current was detected, as well as a -15 mV right-shift in the voltage dependence of activation (Fig. 7B, C). Despite this effect on steady state current, the $h$-infinity curve shows that channels can fully inactivate (Fig. 7D). Furthermore, there is no evidence of a time-dependent change in the selectivity, as evidenced by the lack of two components in the ionic currents (Supplementary Fig. 7). On the other hand, when we performed the double alanine mutation on the identified residues in DII S6 (L792A_L796A, DIIAA Fig. 7A), the effects were different to what was described for DI, DIII, and DIV. In DIIAA, the ionic current showed robust and complete fast inactivation across all voltages tested (Fig. 7E). Despite the lack of steady state current, at the end of the depolarizing pulse, a large tail current was observed. The most likely explanation for the origin of these tail currents is that they are gating currents. This correlates with the observation of two distinct current components during the depolarizing pulses (Fig. 7F). One component that is fast and always outward, and another that follows the reversal potential for sodium and is similar to the ionic currents seen in the WT channels. After the external application of TTX, the first component persisted while the second diminished (Fig. 7G). Clearly, the first component originates from gating charge movement, and the second component is ionic current. These results indicate that there exists significant uncoupling between PD and VSD in DIIAA. Altogether, the results of the

DIA and DIIAA, despite their possible relevance, do not show unequivocally that these residues form part of the inactivation gate, unlike DIIIAA and DIVAA.

## Discussion

### The controversies on fast inactivation gate: location and identity

Fast inactivation is one of the defining features in Nav channels and has implications for many physiological processes. The IFM motif has long been considered the inactivation particle that blocks the pore during a ball and chain-type fast inactivation[18,21]. Manipulations to the IFM motif can lead to severe impairments in fast inactivation; for instance, by mutating IFM to QQQ, fast inactivation can be removed completely[18,38]. Moreover, adding a small peptide that contains the IFM sequence into the intracellular side of the membrane can partially rescue the fast inactivation in the QQQ mutant, which suggests IFM could block directly the open pore[39]. However, when the first structure was solved for mammalian Nav channels[22], it clearly contradicted the *status quo* model. Several Nav structures showed not only that the IFM motif was, in principle, far away from the pore in the putative inactivated state but also that some residues previously identified to influence fast inactivation did not interact directly with IFM at all, L437 and A438 being the prime example[40].

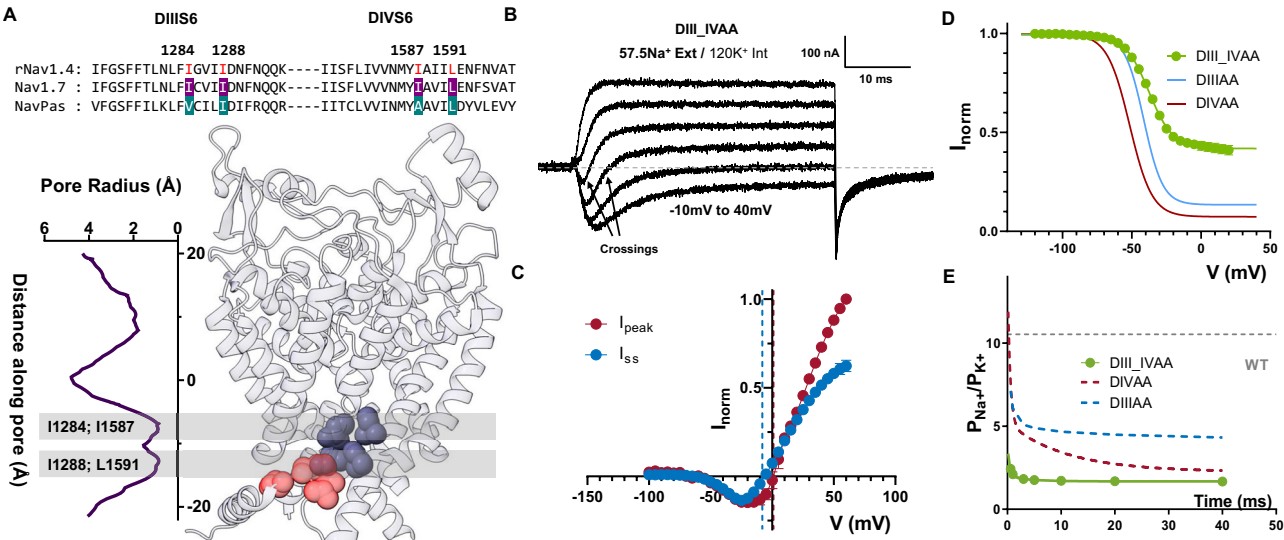

**Fig. 6 | Quadruple alanine mutation of the identified residues in DIII & DIV S6 enhanced the phenotype seen in DIIIAA and DIVAA. A** Sequence alignment of DIII, DIV S6 showing identified residues (highlighted in red) and their positions in the Nav1.7 structure. **B** Representative current traces elicited at different voltages (−10 to 40 mV) for the quadruple alanine mutation (I1284A_I1288A_I1587A_L1591A, DIII_IVAA) in 57.5 mM external Na⁺ and 120 mM internal K⁺. **C** *I−V* curves for the peak (red, *N* = 4) and steady state (blue, *N* = 4) current. The reversal potential for each component is denoted with a corresponding dashed line. **D** *h−*infinity curve for DIII_IVAA (green), DIIIAA (red), and DIVAA (blue). **E** Relative Na⁺/K⁺ permeability vs. time for DIII_IVAA (57.5 mM Na⁺ external and 120 mM K⁺ internal). Dashed gray, blue, and red lines show the WT, DIIIAA and DIVAA permeability, respectively. The solid green line represents an exponential fit to obtain the time constant. Data are shown as mean ± SEM.

An important "missing link" in the Nav fast inactivation saga relates to the fact that, while there are multiple high-resolution Nav cryoEM structures, actually assigning a precise functional state to individual structures is among the current major challenges in structural biology. While it is possible that the structures were resolved in the fast inactivated state, it is also possible that they may have been captured in a slow inactivated state, a different process from fast inactivation[37,41] with putatively significantly different structural features. Yet, if we assume that some of the existing structures represent a fully or partially fast inactivated conformation, the S6 helices, and associated bulky hydrophobic side chains logically emerge as strong candidates to block ion flow during fast inactivation (Fig. 1).

It is worth pointing out that the different Nav structures do not offer a clear-cut consensus regarding the exact residues that could form the fast inactivation gate. Different S6 residues have been proposed based on the different structures (Nav1.5[33] and Nav1.7[23,28]). However, detergents were found lodging at the bundle crossing region of many Navs, potentially distorting the conformation of the protein around that region and further clouding potential interpretations. Here, our experiments and interpretations are based on two structures that are free from this problem, NavPas and Nav1.7 M11. Even though NavPas has never been functionally characterized and Nav1.7 M11 has 11 stabilizing mutations incorporated, our functional data support the idea that the pore profiles from these two channels are likely to be similar to the ones under physiological conditions and represent the fast inactivated pore. Therefore, they serve as a framework to analyze the fast inactivation gate.

In DIII and DIV, none of the single alanine mutations yielded significant change to the fast inactivation. Only when both residues in the same domain or when quadruple residues were mutated to alanine did we start to see a significant amount of steady state current from a leaky inactivated state (Figs. 2, 5, 6). This result clearly demonstrates that an S6-based fast inactivation gate comprises at least two layers of residues with side chains pointing into the pore. The requirement of two layers to make the gate might also

be the reason why many previous single alanine scanning studies on S6 did not show a clear effect and therefore were not able to identify the fast inactivation gate[42]: the two rings of identified bulky hydrophobic residues act *in series*.

Our results show that in DIIIAA and DIVAA, the channels go through all the conformational changes along the fast inactivation pathway. This is supported by several lines of evidence: (i) the kinetics of the residual fast inactivation are similar to the WT; (ii) gating charge immobilization, a process closely linked to fast inactivation, occurs in DIIIAA and DIVAA at the same level and with the same kinetics as the WT; (iii) the slowing down of the tail currents and the change in selectivity, associated with the leaky inactivated state, have the same time course as inactivation; finally (iv) the removal of the fast inactivation process in the IQM mutant relieves the effects observed in DIIIAA. The most direct interpretation of these results is that the double alanine mutations DIIIAA and DIVAA did not affect the sequence of conformational changes that lead to inactivation but rather make the inactivated state conductive.

## Potential coupling between DIII, DIV S6 helices and the SF

An unexpected result from our work was that the ionic selectivity changed concomitantly following the kinetics of fast inactivation in DIIIAA and DIVAA. Na⁺ ions were more permeable than K⁺ ions during the early component (before fast inactivation set in), while the relative K⁺ permeability increased over time. Given that TTX blocks both components of the current completely, Na⁺ and K⁺ are clearly conducted through the pore domain of the channel. It is worth noting that the general behavior of DIIIAA and DIVAA show a striking resemblance with that of batrachotoxin (BTX) modified Nav channels. BTX modifies Nav channels in three major ways: (i) it removes fast inactivation; (ii) it shifts the *G−V* curve to the left; (iii) it alters the selectivity of the channel[43]. Parallels could be found in DIIIAA and DIVAA, corresponding to all three modifications. More intriguingly, BTX is hypothesized to bind to the neurotoxin receptor site 2, which is close to the mutated residues (I433, N434, L437 in DI S6 and F1579 N1584 in DIV S6 have been shown to be crucial for BTX binding)[44]. Therefore, it is likely that

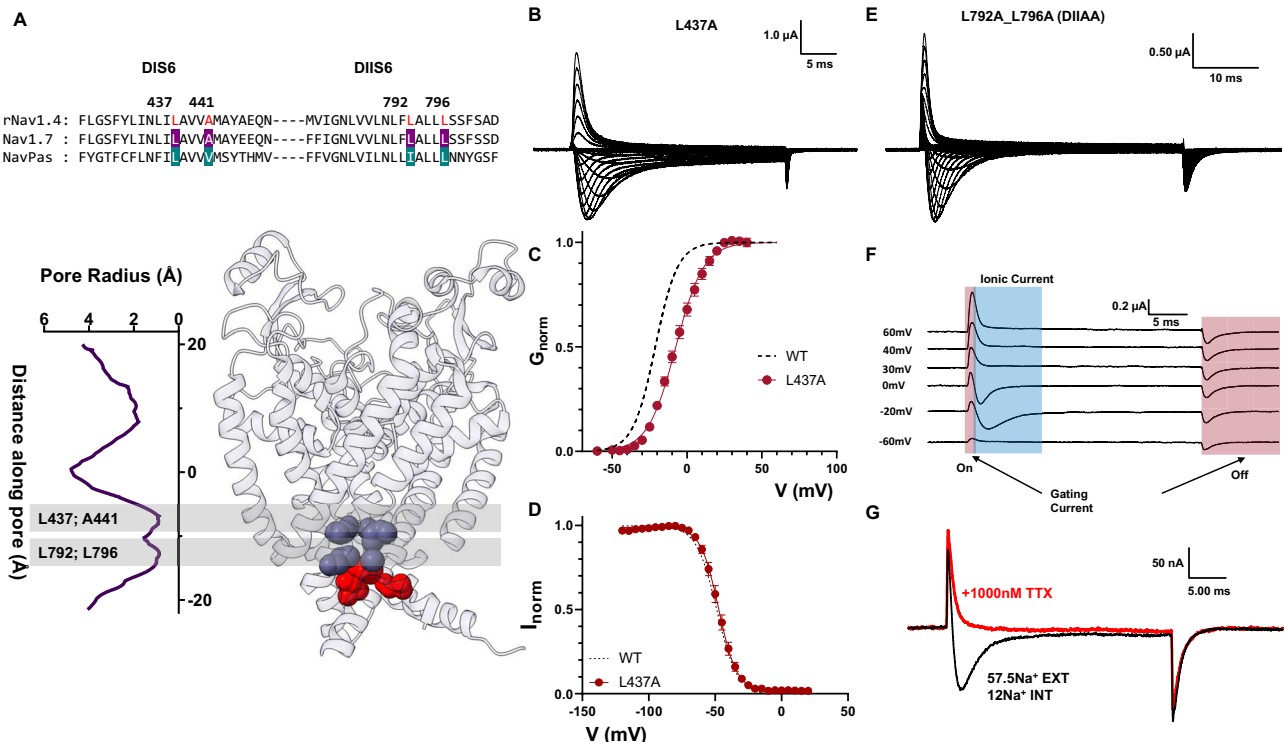

**Fig. 7 | Effects of alanine mutations in DI and DII S6 are different from the ones in DIII and DIV S6. A** Top: sequence alignment of DI and DII S6 showing identified residues (L437 and A441 in DI, L792 and L796 in DII, highlighted in red) and their positions in the Nav1.7 structure. **B** Representative ionic traces for L437A. The ionic conditions used were 57.5 mM Na⁺ outside and 12 mM Na⁺ inside. **C** G–V curves for WT (black dashed line, N = 4) and L437A (red, N = 8). G–V curves were calculated from the peak. **D** h–infinity curve for WT (black, N = 4) and L437A (red, N = 4). **E** Representative current traces for L792A_L796A (DIIAA). The ionic conditions used were 57.5 mM Na⁺ outside and 12 mM Na⁺ inside. **F** Details of DIIAA gating and ionic current components. The red and blue shadowing indicates the presence of gating and ionic currents, respectively. **G** Current from DIIAA collected before (black trace) and after (red) external application of TTX. Data are shown as mean ± SEM.

they share a similar underlying mechanism so BTX evolved as a toxin to disrupt the inactivation gate at S6 even more effectively than DIIIAA and DIVAA.

**Asymmetry in the Nav channel pore**

Due to the asymmetrical assembly of Nav channels, each domain is only similar, not identical to the others, each one playing slightly different roles in the activation and inactivation processes[34,45]. In this work, we also observed the effects of this asymmetry. In DI, only one of the identified residues was bulky and hydrophobic (L437), while the second position was an alanine (A441). L437A produces only small changes to fast inactivation. The apparent conclusion would be that either the residues at DI S6 did not contribute enough to the fast inactivation or our analysis of the structures was not able to identify the correct residues in DI for the fast inactivation gate. However, we cannot rule out the importance of residue L437 since previous work has demonstrated the double L437C/A438W mutant (CW) removes close to 90% of Nav fast inactivation[40]. Our data show that residues located at DI S6 are able to influence fast inactivation, however, the effects observed are not sufficient to unequivocally assign it as part of the fast inactivation gate.

The DII S6, double alanine (DIIAA) seems to not be involved in the inactivation gate. We demonstrated that in DIIAA the gating current became disproportionally larger compared to the ionic current, therefore, it is likely that the double alanine mutation in DII S6 decreases the open probability of the channel. DII VSD has been shown to be involved mostly in channel activation instead of fast inactivation[45]. We suspect that this apparent decrease in P_O was a result of an uncoupling between the VSD and the PD during activation. Our results suggest that the bottom of the S6 region in DII plays a different role from the other domains. Even though our data support an uncoupling between VSD and PD, we cannot rule out the possibility of a drastic enhancement of inactivation similar to the effects of W434F mutation in Shaker K⁺ channel[46,47]. Therefore, it requires further investigation to determine the underlying mechanism of this uncoupling.

**Fast inactivation as a multi-step process**

Our results show that the bottom of the S6 region of the pore serves as the fast inactivation gate. One implication of these results is that fast inactivation is a multi-step process, and the IFM motif binding is only one step, albeit critical, in a whole sequence of conformational changes. Previously, the fast inactivation process was largely described as a two-step process: VSD in DIV activates, exposing the binding pocket for the IFM motif, and IFM motif binds, blocking the pore. Our experiments provide a different interpretation of the mechanism of fast inactivation in Nav channels as a series of conformational changes. In this model (Fig. 8), the activation of VSD in DIV triggers the movement of the IFM motif by exposing a binding pocket. The IFM motif then binds to its binding pocket, which is away from the pore. The binding site of the IFM is far from the S6 regions, therefore, it is expected that the binding event of the IFM is transduced to the S6 gate in the pore through a yet-to-be-defined pathway. Once the movement is allosterically relayed to the S6 segments, the large residues at the bottom of S6 occlude the pore and stop the permeation. The exact nature of these conformational changes and the residues involved in the third step is still elusive, and the underlying mechanism is yet to be determined. However, either a rotation, translocation, or slippage of the S6 helices towards the pore could lead to pore closure during fast inactivation.

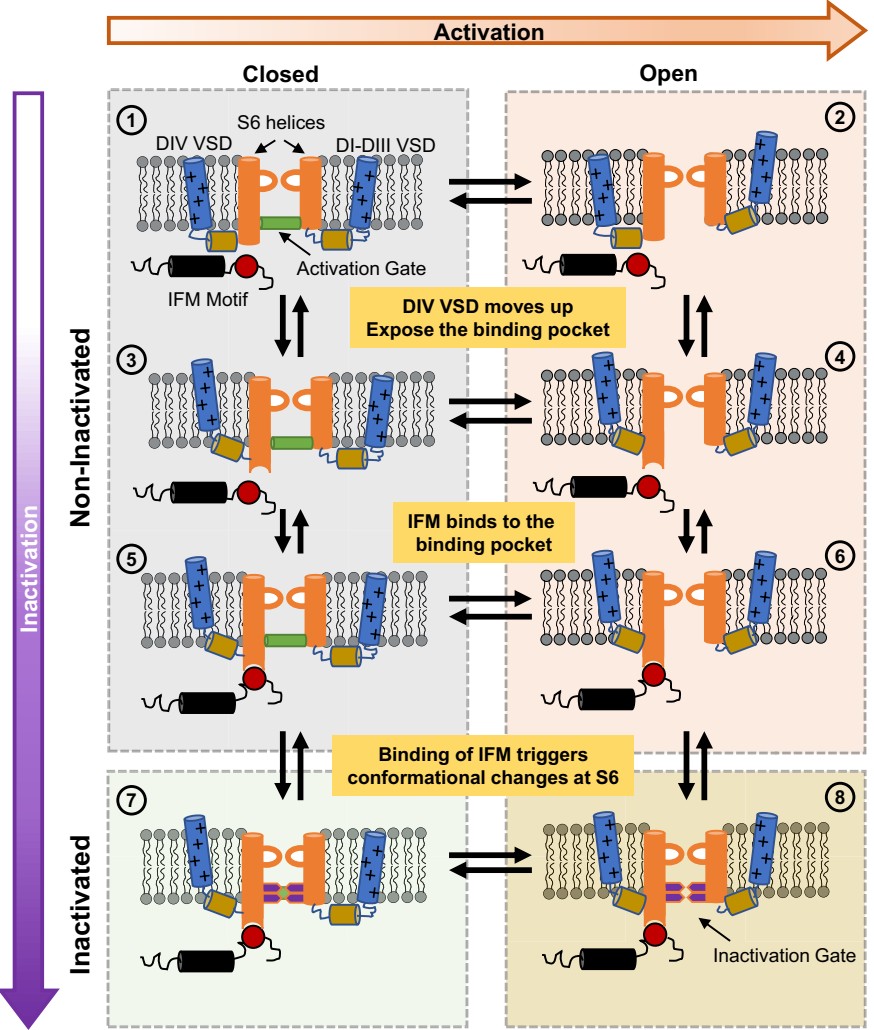

**Fig. 8 | A general model for fast inactivation from open or closed states.** For simplicity, the multi-step activation is shown as a single transition between the last closed state and the open state in the horizontal direction (rightward), and vertical transitions represent the fast inactivation pathway (downward). ① represents the Nav channel in the last closed state. ② represents the channel with the VSD of DI–DIII in the active (up) position, DIV VSD in the resting (down) position, and a conducting pore: an open channel. Transitions from ① to ③ and ② to ④ occur when DIV VSD moves up, exposing the binding pocket for the IFM motif outside the pore region. Once the binding pocket becomes available, the IFM motif binds and transitions from ③ to ⑤ and ④ to ⑥ occur. The binding of the IFM motif triggers conformational changes that are conducted to the pore region closing the two-layered fast inactivation gate, leading to the closed inactivated state in ⑦ or the open inactivated state in ⑧. Please note that DIII VSD is also involved, and this diagram aims to provide a simplified general model for the fast inactivation process.

Finally, our results show a change in selectivity during the leaky inactivated state, which implies a previously unrecognized communication between S6 and the SF. This interaction opens the intriguing possibility that during the normal inactivated state, the sodium channel is less selective to Na⁺. Regardless, it is clear that in light of the current work, many previously identified mutations that influence fast inactivation might require reinterpretation.

## Methods
### Structure analysis and pore radius calculation
Deposits of structures and electron density maps were analyzed using ChimeraX[48] and VMD[49]. The pore radius was calculated by HOLE[31]. Sequencing alignment was performed using SnapGene.

### Site-directed mutagenesis and RNA synthesis
Rat Nav1.4 α and β1 subunit cloned into pBSTA vector, flanked by β-globin sequences, were used[34]. Mutagenesis was performed utilizing mismatch mutagenesis primers in a two-staged PCR reaction for best efficiency (Stratagene). The PCR products were used to transform *Escherichia coli* XL-gold competent cells (Agilent). After ampicillin resistance screening, plasmids were purified using a standard DNA miniprep (Macherey-Nagel). Each DNA was sequenced to confirm the desired mutation and absence of off-target mutations. DNAs were linearized at the unique NotI-HF restriction site (New England Biolabs), and then transcribed into complementary RNA using T7 in-vitro transcription kits (Ambion).

### Channel expression in *Xenopus* oocytes
Ovaries of *Xenopus laevis* were purchased from XENOPUS1. The follicular membrane was digested by collagenase type II (Worthington Biochemical Corporation) 2 mg/ml supplemented with bovine serum albumin (BSA) 1 mg/ml. After defolliculation, Stage V–VI oocytes were microinjected with 50–150 ng of premixed cRNA with 1:1 molar ratio of α and β1 subunits. Injected oocytes were incubated at 18 °C for 1-5 days prior to recording in SOS solution (in mM: 100 NaCl, 5 KCl, 2 CaCl₂, 0.1 EDTA, and 10 HEPES at pH 7.4) supplemented with 50 µg/mL genta-mycin. Unless otherwise stated, chemicals were purchased from Sigma-Aldrich.

## Electrophysiology

Ionic and gating currents were recorded using the cut-open voltage–clamp technique[50]. Micropipettes with resistance between 0.3 and 0.8 MΩ were used to measure the internal voltage of the oocytes. Current data were filtered at 20 kHz with a low-pass Bessel filter online and subsequently sampled by a 16-bit analog-to-digital converter at 1 MHz. Transient capacitive currents were first compensated by a dedicated circuit and then minimized by an online P/N protocol[51]. Temperature was maintained at 11.5 ± 1 °C by a feedback Peltier device throughout the experiment. To lower the series resistance, 1 M cesium methylsulfonate (MES) solution was used in the shortened agar bridges. For ionic current experiments, external solutions consisted of in mM: 57.5 NaMES, 62.5 N-methyl-ᴅ-glucamine (NMG) MES, 2 CaMES, 10 HEPES, and 0.1 EDTA, pH = 7.4, except for cases where the sodium concentration is different, as indicated in the figures; Sodium internal solutions consisted of in mM: 12 NaMES, 108 NMG MES, 10 HEPES, and 2 EGTA, pH = 7.4; Potassium internal solution consisted of in mM: 120 KMES, 10 HEPES, and 2 EGTA, pH = 7.4. Subtraction voltage was set between −80 mV to −120 mV with a P/−4 protocol for ionic currents. Gating currents were recorded in mM: 120 NMG MES, 2 CaMES, 10 HEPES, 0.1 EDTA external solution, and 120 NMG MES, 10 HEPES, 2 EGTA internal solution. Both solutions were titrated to pH 7.40. To block ionic conductance, 750 nM of tetrodotoxin (TTX) was added to the external bath solution. Subtraction voltage was set at +20 mV with a P/4 protocol. For records containing both ionic and gating currents, the recording solutions were consistent with the solutions used for the ionic current experiment. To minimize the loss of gating currents due to the P/−4 protocol while maintaining good resolution of ionic currents and good health of the oocytes, −130 mV was chosen to be the subtraction voltage. Voltage–clamp speed, measured by capacitive transients, yielded a time constant of around 75 μs, and the kinetics of the gating current was reliably resolved.

## Internal solution dialysis for oocytes

To ensure the ionic composition of the oocyte cytoplasm, we exchanged the internal solution in oocytes by incubation in a divalent free solution made up of in mM: 120 KCl, 10 HEPES, 0.1 EDTA and 2 EGTA, pH = 7.4. The oocytes were incubated in the dialysis solution between 30 min to 1 h at room temperature prior to experiments.

## Data analysis

GraphPad 9 (Prism), Excel (Microsoft), Matlab R2022a (Mathworks), and in-house software (Analysis and GPatchM) were used to process all the results.

(I) Steady state fast inactivation curve (h−infinity curve) was assembled by plotting normalized peak Na⁺ current during test pulse against the conditioning pulse voltage and fitted using a two-state model,

$$FInact(V_{CP}) = 1 - \frac{1 - Base}{1 + e^{-z_I F(V_{CP} - V_{I\_1/2})/RT}} \quad (1)$$

Where Finact ($V_{CP}$) is the normalized remaining current after a given conditioning pulse at voltage $V_{CP}$. Base, $z_I$ and $V_{I\_1/2}$ are the percent of non-inactivating current, the apparent charge of the process, and the half-inactivation voltage, respectively. $R$, $T$, and $F$ represent typical values for the gas constant, experiment temperature, and Faraday constant, respectively.

(II) To determine the percentage of charge immobilization, the off-gating current was fitted by a two-component exponential decay.

$$I_{Off-Gating}(t) = A_{Fast} e^{-\frac{t}{\tau_{Fast}}} + A_{Slow} e^{-\frac{t}{\tau_{Slow}}} \quad (2)$$

Where $A_{Fast}$, $\tau_{Fast}$, $A_{Slow}$, and $\tau_{Slow}$ represent amplitudes and time constants of the fast and slow components, respectively.

The amount of charge in the slow component was calculated by multiplying $A_{Slow}$ and $\tau_{Slow}$. The percentage of charge immobilization was then determined by dividing the amount of charge in the slow component by the total amount of off-gating charge, as is shown

$$ChargeImm(t_{CP}) = \frac{A_{Slow} \times \tau_{Slow}}{\int_{t=0}^{t\to\infty} I_{Off-Gating} dt} \quad (3)$$

Where ChargeImm($t_{CP}$) is the percentage of charge immobilization after a conditioning pulse of duration $t_{CP}$ and $\int_{t=0}^{t\to\infty} I_{Off-Gating} dt$ is the total gating charge as measured by the integral of the off-gating current. In some cases, the total gating charge is calculated by summing the slow component and fast component.

(III) The time constant of inactivation or deactivation kinetics were calculated by fitting the ionic currents with either one or two exponential decays using the general equation

$$I(t) = A_1 e^{-\frac{t}{\tau_1}} + A_2 e^{-\frac{t}{\tau_2}} + I_{ss} \quad (4)$$

Where $A_1$, $\tau_1$, $A_2$, and $\tau_2$ represent amplitudes and time constants of the first and second components, respectively. $I_{SS}$ is the steady state current. When one exponential was used, the term of the second exponential component was eliminated.

To obtain the weighted time constant, we used the following relationship

$$\tau_W = \frac{A_1 \tau_1 {}^* A_2 \tau_2}{A_1 + A_2} \quad (5)$$

(IV) Relative permeability between Na⁺ and K⁺ ions was determined in a bi-ionic environment with 120 mM intracellular K⁺ and different extracellular Na⁺ concentrations. The reversal potential was determined by fitting the instantaneous I–V curve with a linear relationship and locating the intersection point on the voltage axis. The reversal potentials as a function of depolarization time were subsequently used to determine the relative permeability using (Eq. 6), derived from GHK equation[52,53].

$$V_{rev}(t) = \frac{RT}{F} ln\left(\frac{P_{Na}[Na^+]_o}{P_K[K^+]_i}\right) \quad (6)$$

where $V_{rev}(t)$ is the reversal potential after depolarizing pulse of duration $t$; $P_{Na}$ and $P_K$ represent relative permeability of Na⁺ and K⁺, respectively; $[Na^+]_o$ and $[K^+]_i$ represent extracellular Na⁺ concentration and intracellular K⁺ concentration, respectively.

(V) The ionic conductance ($G(V_m)$) was calculated by dividing either the peak current or steady state current by the experimentally determined driving force at each depolarizing voltage ($V_m$). Subsequently, the curve was normalized to the maximal conductance across all voltages ($G$max) to obtain the conductance versus voltage (G–V) curves. The G–V curves were fitted using a two-state model:

$$\frac{G(V_m)}{G \max} = \frac{1}{1 + e^{-z_G F(V_m - V_{G\_1/2})/RT}} \quad (7)$$

Where $V_{G\_1/2}$ is the half activation voltage, and $z_G$ is the apparent charge.

(VI) Total gating charge movement during voltage pulses was measured by integrating the gating currents. Charge–voltage (Q–V) curves were obtained by plotting the normalized total charge movement ($Q_{Norm}$) at each depolarizing voltage and fitted using a two-state

model (Eq 8),

$$Q_{\text{Norm}}(V_m) = \frac{1}{1 + e^{-z_Q F(V_m - V_{Q\_1/2})/\text{RT}}} \qquad (8)$$

Where $V_{Q\,1/2}$ is the gating half activation voltage, and $z_Q$ is the apparent charge.

## Reporting summary
Further information on research design is available in the Nature Portfolio Reporting Summary linked to this article.

## Data availability
All data and resources presented in the paper will be made available to readers upon request. The protein structures used in this work can be accessed with the following accession codes: 6A95, 7XVF, and 6UZ3. Source Data are available as a Source Data File or via the Figshare repository. Source data are provided in this paper.

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

## Acknowledgements

We would like to thank Dr. Eduardo Perozo and Dr. Chris Ahern for their thoughtful comments and careful editing of the manuscript; Dr. Rong Shen for his help in calculating the pore radius in the cryo-EM structures. This work is supported by the National Institutes of Health Award R01GM030376, F.B. and National Science Foundation Award QuBBE QLCI (NSF OMA-2121044), F.B. B.P. is a PEW Latin American Fellow (2019).

## Author contributions

Conceptualization: Y.L. and F.B. Performed and analyzed experiments: Y.L. Interpreted results: Y.L., C.B., B.P., and F.B. Writing: Y.L., C.B., B.P., and F.B. Supervision: F.B.

## Competing interests

The authors declare no competing interests.
