## [Peer Review File · Nature Communications]

A Mechanistic Reinterpretation of Fast Inactivation in Voltage-Gated Na⁺ ChannelsReviewers' Comments:

Reviewer #1:

Remarks to the Author:

This is an important paper. The cryo-EM structures of vertebrate voltage-dependent sodium channels have shown that the long-held model of channel inactivation as physical occlusion of the pore by the IFM motif in the III-IV intracellular linker is not correct, because the IFM motif was not in the pore but rather buried in a pocket fairly distant from the pore. However, the structures do not give much insight into exactly how binding of the IFM motif in this pocket produces a closed, inactivated channel. Here, the authors use the structures of inactivated channels to recognize two rings of hydrophobic residues that form a "double-seal" near the inner mouth of the pore. Using this modeled structure, they then show that mutating a residue in each ring results in an inactivated channel that is no longer fully closed but instead is leaky, thus offering strong support for their proposed mechanism. This is probably the single most remarkable result from mutagenesis of an ion channel that I know, because it would have been impossible to come up with this combination of mutations without thinking deeply about the structure. Even more remarkably, the authors find that the leaky channel formed by these mutations has an altered ionic selectivity, with reduced selectivity for Na over K. This implies that there is some sort of coupling between this region near the inner pore and the selectivity filter of the channel located near the outer pore.

Besides presenting fascinating and very important results, the paper is a technical tour-de-force of channel biophysics, using well-designed measurements of gating kinetics, gating current, and ionic selectivity in the mutated channels to lead the authors to their conclusions and new model. The presentation of the motivation for the experiments and the discussion of their interpretation are done with clarity and concision. I can find only minor points of wording that might be improved.

Minor

13 "The hinged-lid model is long accepted as the canonical model for fast inactivation" would be accurate as "13 The hinged-lid model was long accepted...." . It has now been 6 years that the cryo-EM structures showed that the model is incorrect.

44 "positively charge" should be "positively charged"

57 "Upon the first mammalian Nav channel structure (22) it became clear that the IFM motif, though docking into a hydrophobic pocket, resided far away from the pore in the putative inactivated state to block the permeation path, in contrast to the predictions of the canonical "hinged-lid" model."

I found this sentence a little difficult to interpret. Maybe the meaning would be clearer as "Upon the first mammalian Nav channel structure (22) it became clear that the IFM motif docked into a hydrophobic pocket far away from the pore rather than physically occluding the pore, in contrast to the predictions of the canonical "hinged-lid" model."

104 "Given the fact that both structures are determined at 0mV and the DIV VSD is in the "up" conformation, it is reasonable to assume that the pore resemble a fast inactivated state."

Since channels in cryoEM have been at 0 mV for a long time, I would think the structure is more likely to be a slow inactivated state rather than fast inactivated state. It does not seem necessary or useful to get into a discussion of this in the present paper, but the sentence might be slightly more accurate as "Given the fact that both structures are determined at 0mV and the DIV VSD is in the "up" conformation, it is reasonable to assume that the pore is in an inactivated state."

439 "In DIII and DIV, none of the single alanine mutation yielded..." should be "mutations"

Reviewer #2:

Remarks to the Author:

Liu et al investigate whether hydrophobic residues lining the intracellular pore are part of the inactivation gate of Nav channels. Given that Nav channel inactivation is essential to the function of excitable cells, insight into this mechanism is of high interest. Previous work suggests that an IFM motif located within the intracellular DIII-DIV linker of Nav channels directly blocks the channel pore to induce inactivation. However, recent Nav channel structures show that the binding of the IFM motif

is far from the pore, even though it is required for inactivation. The authors demonstrate that inactivation of the channel is impaired by mutation of several intracellular pore residues on the DIII and DIV S6 segments, preventing inactivation even though IFM is still intact, suggesting that these residues are part of the inactivation mechanism.

The manuscript is well-written and delivers a hypothesis for a new mechanism for the fast inactivation voltage-gated sodium channels. The experimental results shown provide evidence for the connection of intracellular S6 helices to an inactivation gate.

Major Comments:

- Voltage clamp fluorometry is likely to provide higher resolution data on whether the VSDs are being affected, including changes in activation and deactivation kinetics. It is surprising that these experiments were not included.
- The addition of the IQM to DIIIAA was helpful in assessing how these different channel components interact. Are the results with IQM added to DIVAA similar?
- The model proposed in Figure 8 is very DIV centric even though the data presented in this work and previous work by this group and others have shown a significant role for DIII.
- The authors provide a comparison between different species to demonstrate the conservation of the amino acids observed in this study. It would be valuable to compare across Nav channel isoforms that inactivate if the mechanism is thought to be universal, as suggested.
- Some caveats should be included. Without identifying the coupling pathway, it is possible that the mutation of the S6 residues interrupts the normal inactivation pathway even though these positions might not directly participate. Further work is needed to convincingly make definitive statements about the participation of these positions in the inactivation gate. As noted in the manuscript and shown in refs 40 and 42, DI mutations have been shown to have similar effects on inactivation.

Minor Comments

- The different treatment of showing DIA and DIIAA data in figure 7 is confusing. Why not plot DIIAA GV and inactivation curves along with DIA?
- It would be helpful to include hypotheses relating to the coupling of S6 helices to IFM binding that would cause inactivation via the intracellular S6 residues that were identified.

RESPONSE TO THE REVIEWERS' COMMENTS

We thank the reviewers for their insightful comments. The modifications they proposed have made the manuscript much better. All the changes in the text are marked in yellow and they include the changes as a result of the reviewers' comments, the changes required by the Editor, in addition to other small corrections. We also include an extra Supplementary File with changes in yellow and the file is labeled MARK.

Reviewer #1 (Remarks to the Author):

This is an important paper. The cryo-EM structures of vertebrate voltage-dependent sodium channels have shown that the long-held model of channel inactivation as physical occlusion of the pore by the IFM motif in the III-IV intracellular linker is not correct, because the IFM motif was not in the pore but rather buried in a pocket fairly distant from the pore. However, the structures do not give much insight into exactly how binding of the IFM motif in this pocket produces a closed, inactivated channel. Here, the authors use the structures of inactivated channels to recognize two rings of hydrophobic residues that form a "double-seal" near the inner mouth of the pore. Using this modeled structure, they then show that mutating a residue in each ring results in an inactivated channel that is no longer fully closed but instead is leaky, thus offering strong support for their proposed mechanism. This is probably the single most remarkable result from mutagenesis of an ion channel that I know, because it would have been impossible to come up with this combination of mutations without thinking deeply about the structure. Even more remarkably, the authors find that the leaky channel formed by these mutations has an altered ionic selectivity, with reduced selectivity for Na over K. This implies that there is some sort of coupling between this region near the inner pore and the selectivity filter of the channel located near the outer pore.

Besides presenting fascinating and very important results, the paper is a technical tour-de-force of channel biophysics, using well-designed measurements of gating kinetics, gating current, and ionic selectivity in the mutated channels to lead the authors to their conclusions and new model. The presentation of the motivation for the experiments and the discussion of their interpretation are done with clarity and concision. I can find only minor points of wording that might be improved.

We appreciate the reviewer positive feedback and the recognition of the importance of our findings.

Minor

13 "The hinged-lid model is long accepted as the canonical model for fast inactivation" would be accurate as "13 The hinged-lid model was long accepted..." . It has now been 6 years that the cryo-EM structures showed that the model is incorrect.

We thank the reviewer for spotting this typo and we have changed it in the manuscript.

44 "positively charge" should be "positively charged"

We thank the reviewer for spotting this and we have changed it in the manuscript.

57 "Upon the first mammalian Nav channel structure (22) it became clear that the IFM motif, though docking into a hydrophobic pocket, resided far away from the pore in the putative inactivated state to block the permeation path, in contrast to the predictions of the canonical "hinged-lid" model."

I found this sentence a little difficult to interpret. Maybe the meaning would be clearer as "Upon the first mammalian Nav channel structure (22) it became clear that the IFM motif docked into a hydrophobic pocket far away from the pore rather than physically occluding the pore, in contrast to the predictions of the canonical "hinged-lid" model."

We agree with the reviewer that by stating that particular sentence in the suggested way makes it clearer to understand. We have rephrased this particular sentence in the manuscript.

104 “Given the fact that both structures are determined at 0mV and the DIV VSD is in the “up” conformation, it is reasonable to assume that the pore resemble a fast inactivated state.”

Since channels in cryoEM have been at 0 mV for a long time, I would think the structure is more likely to be a slow inactivated state rather than fast inactivated state. It does not seem necessary or useful to get into a discussion of this in the present paper, but the sentence might be slightly more accurate as “Given the fact that both structures are determined at 0mV and the DIV VSD is in the “up” conformation, it is reasonable to assume that the pore is in an inactivated state.”

We agree with the reviewer that it is not necessary to discuss the exact state the structures were captured in. We have rephrased that particular sentence as suggested.

439 “In DIII and DIV, none of the single alanine mutation yielded...” should be “mutations”

We thank the reviewer for spotting this typo and we have changed it in the manuscript.

Reviewer #2 (Remarks to the Author):

Liu et al investigate whether hydrophobic residues lining the intracellular pore are part of the inactivation gate of Nav channels. Given that Nav channel inactivation is essential to the function of excitable cells, insight into this mechanism is of high interest. Previous work suggests that an IFM motif located within the intracellular DIII-DIV linker of Nav channels directly blocks the channel pore to induce inactivation. However, recent Nav channel structures show that the binding of the IFM motif is far from the pore, even though it is required for inactivation. The authors demonstrate that inactivation of the channel is impaired by mutation of several intracellular pore residues on the DIII and DIV S6 segments, preventing inactivation even though IFM is still intact, suggesting that these residues are part of the inactivation mechanism.

The manuscript is well-written and delivers a hypothesis for a new mechanism for the fast inactivation voltage-gated sodium channels. The experimental results shown provide evidence for the connection of intracellular S6 helices to an inactivation gate.

We thank the reviewer for the thorough evaluation of our manuscript.

Major Comments:

- Voltage clamp fluorometry is likely to provide higher resolution data on whether the VSDs are being affected, including changes in activation and deactivation kinetics. It is surprising that these experiments were not included.

We believe gating current measurement is more suitable for this particular work because it allows us to directly assess charge immobilization, which is a critical aspect of inactivation. Additionally, by obtaining robust gating current recordings in all mutants with a *high signal-to-noise ratio* (which is not the case with fluorometry), we can accurately analyze the kinetic properties. While site-directed voltage clamp fluorometry (presently ongoing) may provide valuable information on individual VSD movements, our focus in this manuscript is primarily on understanding the relationship between the IFM motif and the inactivation gate. As such, measuring gating currents aligns with the traditional approach for characterizing charge immobilization and enables a comprehensive analysis of the inactivation mechanism.

- The addition of the IQM to DIIIAA was helpful in assessing how these different channel components interact. Are the results with IQM added to DIVAA similar?

Interesting point raised by this reviewer. We did not try IQM_DIVAA because, unfortunately, DIVAA express much less than DIIIAA and IQM also decreases expression significantly.

- The model proposed in Figure 8 is very DIV centric even though the data presented in this work and previous work by this group and others have shown a significant role for DIII.

The reviewer is correct to point the involvement of DIII on inactivation. The model is clearly a simplification but, to correct this oversight, we have added a note in the figure legend indicating that DIII is also participating in inactivation.

- The authors provide a comparison between different species to demonstrate the conservation of the amino acids observed in this study. It would be valuable to compare across Nav channel isoforms that inactivate if the mechanism is thought to be universal, as suggested.

We thank the reviewer for pointing this out. We agree that while comparing amino acid sequence across different species could provide evolutionary insights, it still remains valuable to include a sequence comparison across different isoforms of Nav channels to demonstrate similarities and special adaptations within them. We have included another supplementary figure (SuppleFig. 2B). It is true that not all the identified residues are conserved across all isoforms. However, the property of the amino acids at those positions are relatively conserved.

- Some caveats should be included. Without identifying the coupling pathway, it is possible that the mutation of the S6 residues interrupts the normal inactivation pathway even though these positions might not directly participate. Further work is needed to convincingly make definitive statements about the participation of these positions in the inactivation gate. As noted in the manuscript and shown in refs 40 and 42, DI mutations have been shown to have similar effects on inactivation.

We would like to respectfully argue against this comment. Indeed, mutations in DI have been shown to disrupt fast inactivation. L437C together with A438W (CW) in particular has been shown to remove fast inactivation almost completely. Considering the location of those residues at S6, it is tempting to conclude that those residues might form the fast inactivation gate. However, we have recorded the charge immobilization profile of CW (unpublished results*) and we discovered that it has been significantly altered which is not case for the DIIIAA and DIVAA mutants described here.

We agree that without the complete knowledge of the entire coupling pathway, there exists a possibility that the identified residues might exert their effects by interrupting the pathway. However, we believe that the existence of a leaky inactivated state provides strong evidence that support our interpretation that the S6 residues identified are the fast inactivation gate and we are not actually interrupting the inactivation pathway (as IQM or CW might be acting).

Minor Comments

- The different treatment of showing DIA and DIIAA data in figure 7 is confusing. Why not plot DIIAA GV and inactivation curves along with DIA?

It is challenging to acquire an accurate GV curve for DIIAA. The mixture of gating current and ionic current at the beginning of the trace makes it unfeasible to reliably obtain the peak for the ionic current. While it is possible to record the gating current together with the ionic current first then apply external TTX to record only the gating current subsequently, this method is highly sensitive to small fluctuation within the system (rundown, changes in capacitance, temperature etc.,). Since, the main conclusion of this work rely mostly on results on DIIIAA and DIVAA, we decided to describe the effects of DIIAA qualitatively and leave room for future investigations.

- It would be helpful to include hypotheses relating to the coupling of S6 helices to IFM binding that would cause inactivation via the intracellular S6 residues that were identified.

We thank the reviewer for this comment. Indeed, we think that there is a chain of amino acids that couple the IFM binding to the closing of the inactivation gate. We have hypothesized the residues that form this chain, and it is the subject of our future research, but as we do not have results yet, we believe that to explicitly state it is premature.

*unpublished results have been communicated as an abstract.

Yichen Liu & Francisco Bezanilla

"Charge immobilization in two different fast-inactivation-removed Nav1.4 sodium channel mutants",

Biophysical Journal,

Volume 121, Issue 3, Supplement 1,

2022,

Pages 95a-96a,

ISSN 0006-3495,

<https://doi.org/10.1016/j.bpj.2021.11.2252>.